# Characterization of *Bacillus* Strains from Natural Honeybee Products with High Keratinolytic Activity and Antimicrobial Potential

**DOI:** 10.3390/microorganisms11020456

**Published:** 2023-02-11

**Authors:** Diego Martín-González, Sergio Bordel, Selvin Solis, Jorge Gutierrez-Merino, Fernando Santos-Beneit

**Affiliations:** 1Institute of Sustainable Processes, Dr. Mergelina s/n, 47011 Valladolid, Spain; 2Department of Chemical Engineering and Environmental Technology, School of Industrial Engineering, University of Valladolid, Dr. Mergelina, s/n, 47011 Valladolid, Spain; 3School of Biosciences and Medicine, University of Surrey, Guildford GU2 7XH, UK

**Keywords:** *Bacillus licheniformis*, honey, bee, keratin, antibiotic, lichenicidin, feed, biodegradation

## Abstract

Two efficient feather-degrading bacteria were isolated from honeybee samples and identified as *Bacillus sonorensis* and *Bacillus licheniformis* based on 16S rRNA and genome sequencing. The strains were able to grow on chicken feathers as the sole carbon and nitrogen sources and degraded the feathers in a few days. The highest keratinase activity was detected by the *B. licheniformis* CG1 strain (3800 U × mL^−1^), followed by *B. sonorensis* AB7 (1450 U × mL^−1^). Keratinase from *B. licheniformis* CG1 was shown to be active across a wide range of pH, potentially making this strain advantageous for further industrial applications. All isolates displayed antimicrobial activity against *Micrococcus luteus*; however, only *B. licheniformis* CG1 was able to inhibit the growth of *Mycobacterium smegmatis*. In silico analysis using BAGEL and antiSMASH identified gene clusters associated with the synthesis of non-ribosomal peptide synthetases (NRPS), polyketide synthases (PKSs) and/or ribosomally synthesized and post-translationally modified peptides (RiPPs) in most of the *Bacillus* isolates. *B. licheniformis* CG1, the only strain that inhibited the growth of the mycobacterial strain, contained sequences with 100% similarity to lichenysin (also present in the other isolates) and lichenicidin (only present in the CG1 strain). Both compounds have been described to display antimicrobial activity against distinct bacteria. In summary, in this work, we have isolated a strain (*B. licheniformis* CG1) with promising potential for use in different industrial applications, including animal nutrition, leather processing, detergent formulation and feather degradation.

## 1. Introduction

Around 1 million tons of chicken feathers are produced as waste products annually by the poultry industry around the world [1]. In general, all types of feathers have a protein content of around 90%, which indicates that this waste could have great potential as a source of protein and amino acids for animal feed or many other applications [2]. However, the most abundant protein in feathers is keratin, an insoluble protein that is difficult to degrade due to the presence of hydrogen bonds, disulphide bonds and hydrophobic interactions [3]. Commonly, the discarded feathers are either dumped in landfills or incinerated. Sometimes, feathers are converted to feather meal by steam pressure and/or chemical treatment and utilized on a limited basis as a dietary protein supplement for animal feedstuffs [4]. However, this approach is unusual due to the poor digestibility of keratin by animals and the high cost of the production process. In fact, in its native state, keratin is not degradable by common proteolytic enzymes such as trypsin, pepsin or papain. Therefore, prior to usage, the feathers are steam pressure cooked or chemically treated to make them more digestible [5]. An alternative to these mechanical and chemical approaches is proteolysis of keratin by enzymatic and/or microbiological methods, thereby improving the nutritional value of feather waste by avoiding the destruction of certain amino acids, such as methionine, tryptophan and lysine [6].

Keratin degradation requires a specific class of proteases known as keratinase [4]. Although several microorganisms, including fungi and bacteria, are known to degrade keratin, *Bacillus* species are considered as the most effective keratin degraders [7]. Lin et al. [8] were the first to successfully isolate and characterize a keratinase enzyme from a keratin-degrading bacterium, *Bacillus licheniformis* PWD-1, that they previously isolated. Recently, we have also characterized a keratin-degrading strain that, in addition to feathers, can degrade other type of wastes, such as bioplastics [9]. The strain, initially referred to as *Bacillus pumilus* B12 [10], was reclassified as *Bacillus altitudinis* B12 after genomic analyses [9]. Other strains with keratinolytic activity have also been reported and, in some cases, highlighted as promising candidates for the development of cost-effective and eco-friendly platforms for the conversion of feathers into value-added products, especially in the form of livestock feed [2,5,6,7]. To develop a biotechnological application for that purpose, it would be of great interest to characterize a strain that, in addition to its keratinolytic activity, would exhibit an important antimicrobial potential. This feature would inhibit certain pathogenic microorganisms present in the feathers, thereby decreasing the risk of causing diseases (such as gastroenteritis or tuberculosis) in animals fed with these wastes [11,12,13]. There is currently a global antibiotic resistance crisis. New antibiotics are rarely discovered, and the number of resistant microbes is increasing worldwide.

At present, scientists are searching for new antimicrobial weapons and therapies against microbial infections in natural products and unexplored ecological niches. For decades, soil has been used as the main source for the identification of novel antibiotics. However, there is an urgent need to explore other ecological niches as the screening of soil samples has become overexploited. Interestingly, ecological niches in which symbionts work together to defend themselves by producing bioactive compounds are relatively underexploited. This is the case in beehives. Some authors have found antimicrobial metabolites in honeybee products, such as pollen and honey, where a significant variety of different bacteria derive from plants and bees [14,15]. Honeybees manufacture honey, bee bread (pollen) and royal jelly using secretions from their stomach, salivary glands and hypopharynx glands, and these natural products are continuously exposed to the environment [15]. Honeybees transfer microorganisms vertically (mother to daughter) or horizontally, via trophallaxis (mouth to mouth feeding) and coprophagy (anus to mouth). These microorganisms have a very important function in the digestion of plant polymers and in the preservation of honey and bee pollen by controlling the growth of pathogens [14,15]. Therefore, honeybee products are promising sources for isolating novel antibiotic-producing microorganisms. In previous work, we have shown that raw honeybee products provide antibiotic-producing isolates (*Streptomyces*) and lactic acid bacteria (LAB) with high antimicrobial potential and probiotic features, respectively [14,15]. Among the antibiotic producers isolated, *Streptomyces* species stand out as most interesting, as they are responsible for production of over half of the antibiotics discovered so far [16,17,18]. On the other hand, the isolated LAB (*Lactobacillus* and *Pediococcus*) can potentially exhibit good probiotic properties in animals and/or humans as they are able, among other capabilities, to activate type-I interferon production [15]. In another study, both probiotic and antimicrobial properties have been described for some *Bacillus* strains isolated from stingless bee honey collected across Malaysia [19]. Therefore, the products that honeybees manufacture are a very valuable source for the isolation of antibiotic-producing bacteria. Whether these bacteria derive as symbionts of honeybees or environmental contaminants is a very important question that remains to be elucidated.

The aim of this study was to isolate novel *Bacillus* strains with both high keratinolytic activity to degrade feather wastes in an efficient manner and antimicrobial potential to inhibit major pathogens, such as *Mycobacterium*. To this end, studies were carried out on antibiotic-producing and probiotic strains previously isolated from natural honeybee products collected from 15 apiaries across Southeast England [20]. Genome sequencing analysis of the isolates resulted in the identification of novel strains. Furthermore, phenotypic characterization was performed in a systematic manner to select for the most promising candidate with potential in industrial applications.

## 2. Materials and Methods

### 2.1. Microbial Growth Conditions and Growth Determination

*Bacillus* cells were grown in a mineral salt medium with feathers as the sole carbon and nitrogen sources, with the following composition (g/L): KH_2_PO_4_ (0.7), K_2_HPO_4_ (1.4), MgSO_4_ (0.1), NaCl (0.5), ZnSO_4_·7H_2_0 (0.05), FeSO_4_·7H_2_0 (0.015) and feathers (10). pH of the medium was maintained at 7, nevertheless, growth of cells was also determined across a range of pH (6–9). Chicken feathers provided by a poultry farm in Valladolid, Spain, were first washed with tap water and Triton-X to get rid of any debris, followed by a final cleaning step with distilled water. Then, clean feathers were dried at 60 °C overnight and manually cut into small pieces. Cultures were performed in either 120 mL or 2.2 L bottles, depending on the experiment. In all experiments, cultures of the respective strain grown until the mid-exponential phase (i.e., OD_600_~1.5) were inoculated into the defined media at 2% (*v/v*). The bottles containing the inoculated cultures were hermetically closed with an isoprene rubber and an aluminum crimp seal and incubated at 37 °C with shaking (150 rpm) for several days. Determination of bacterial cell growth was performed by measuring O_2_ consumption and CO_2_ production in the glass bottles using a Gas Chromatograph (Varian Bruker 430-GC, Middelburg, The Netherlands) equipped with a thermal conductivity detector (GC-TCD), as previously described [9]. Abiotic controls (without bacterial inoculum) were also maintained under the same conditions to monitor O_2_ and CO_2_ concentrations in the bottles, which should remain invariable.

### 2.2. Experimental Design, Sampling, DNA Extraction and 16S rRNA Sequencing

Honeybee samples were collected from 15 apiaries across Southeast England. Sample collection took place between mid-June and mid-August and targeted several habitats and soils in 4 different counties, as well as different beehives, some of which were located within the same apiary [20]. Samples were collected directly from honeycomb frames. Ten grams each of bee product sample were collected in sterile swab tubes and containers respectively, which were then immersed in liquid nitrogen and stored at −80 °C. To ensure full representation of the whole beehive, samples were collected from different parts of the honeycomb.

To enrich the presence of *Bacillus* strains in the honeybee products, the samples were incubated at 80 °C for 10 min to select for endospore-forming bacteria. The samples were serially diluted and seeded onto Tryptone Soya Agar (TSA) medium to favor *Bacillus* growth. After incubation, isolates with stronger antimicrobial properties were selected and further analyzed using 16S rRNA sequencing to select strains belonging to the *Bacillus* genus.

DNA extraction was carried out from cultures in Tryptone Soya Broth (TSB) using the EZNA bacterial DNA kit according to the manufacturer’s instructions (Omega Bio-Tek, Doraville, CA, USA) and as previously reported [18]. 16S rRNA gene amplicon sequencing was performed on an Illumina MiSeq sequencer using universal 16S rRNA bacterial primers for V3–V4 regions. Among the isolates, three were found to be *Bacillus sonorensis*, one was *B. licheniformis*, one was *B. subtilis* and one was an unclassified *Bacillus* species that had 100% sequence similarity to *Bacillus* sp. BAB-4886.

### 2.3. Genome Sequencing, Assembly and Annotation

DNA sequencing was performed by MicrobesNG (University of Birmingham, United Kingdom) using the Illumina MiSeq platform as previously described [21]. Briefly, the library was prepared with the 250 NexteraTM XT Library Prep Kit Genome sequencing and the quality of the generated reads was trimmed with Trimmomatic [22]. The generated contigs were assembled from the paired-end reads using Shovill v.1.0.41 with SPAdes 3.13.0 [23] and the resulting genome assemblies were verified by N50 and L50 using Quast v.4.5 [24] and annotated with Prokka v.1.13 [25]. The secretome for each of the strains was inferred by predicting the presence of signal peptides using SignalP-5.0 [26]. Sequencing reads, genome assemblies and metadata have been uploaded onto Genbank in BioProject PRJNA907715, with the next genome accession numbers: JAPQWB000000000 (Z1), JAPQWC000000000 (CG1), JAPQWD000000000 (X1), JAPQWE000000000 (X2), JAPQWF000000000 (AB6) and JAPQWG000000000 (AB7).

### 2.4. Taxonomic Analysis

For the taxonomy analysis, the web-based tool JSpeciesWS [27] was employed (https://jspecies.ribohost.com/; accessed on 3 November 2022). An initial search for the closest relative of each of the isolates was carried out using Tetra Correlation Search (TCS) [28], followed by a more systematic comparison with *B. subtilis*, *B. sonorensis* and *B. licheniformis* species using pairwise Average Nucleotide Identity with BLAST (ANIb) calculations [29]. The phylogenetic analysis was performed using the dendrogram function in SciPy. A distance between species was defined as 100 minus the obtained ANI values and the resulting matrix of distances was used to construct the dendrogram.

### 2.5. Keratinase Activity Detection Assay

Keratinase activity was determined following previous existing methods [8,30]. Liquid samples of 0.7 mL were daily extracted of each culture and centrifuged 20 min at 13,000 rpm to obtain a final 0.5 mL aliquot of supernatant. With the centrifugation step, cells and feather debris are removed. The 0.5 mL aliquot was mixed with 0.5 mL of 100 mM glycine-NaOH (pH 10) containing 1% casein and incubated at 37 °C for 20 min. In this reaction, tyrosine is released after the breaking of casein by the keratinases, among other enzymes. The reaction was stopped by addition of 0.5 mL of trichloroacetic acid (20% *w/v*) and incubation for 15 min at room temperature. Following the inactivation of the enzymes, the samples were centrifuged during 15 min at 13,000 rpm and the supernatant collected and measured with a spectrophotometer at 280 nm. A standard curve was performed using solutions of 0–700 mg L^−1^ of tyrosine. One keratinase unit (U) was defined as the amount of enzyme required to increase the absorbance by 0.01 (OD_280nm_) in one minute under the assay conditions employed.

### 2.6. Growth Inhibition Bioassay

To study the growth inhibition of *Micrococcus luteus* and *Mycobacterium smegmatis,* aliquots of 5 µL of each of the *Bacillus* isolates were added onto the surface of TSA plates containing 10^8^ CFU × mL^−1^ of the respective indicator strain. Plates were incubated at 30 °C until visible halos of inhibition appeared.

### 2.7. Analysis of Gene Clusters with Antimicrobial Potential

The genomes of *Bacillus* isolates were analyzed to identify gene clusters associated with the biosynthesis of antimicrobial compounds. The secondary metabolites clusters were identified using antiSMASH v.6.0 [31] and BAGEL v.3 [32].

## 3. Results

### 3.1. Isolation and Selection of Bacillus Strains from Raw Honeybee Products

Natural *Bacillus* strains with antimicrobial potential were isolated using a screening approach from samples of raw honey, bee bread (referred as to pollen henceforth) and royal jelly collected from 15 apiaries across Southeast England [20]. Colonies with stronger antimicrobial properties against *Micrococcus luteus* ATCC4698 were selected and further analyzed using 16S rRNA sequencing to select for *Bacillus* species. We isolated six different *Bacillus* strains (Z1, CG1, AB6, AB7, X1 and X2) from either raw honey or royal jelly samples (see Table 1).

### 3.2. Taxonomy of the Bacillus Isolates and General Features of Their Genomes

The 16S rRNA sequencing analysis informed that the six isolates were *Bacillus*, and more specifically *B. subtilis*, *B. licheniformis* and *B. sonorensis*, depending on the strain (see Table 1). To have a better insight about these strains, their genomes were sequenced. The sequenced genomes were used to classify the isolated strains based on whole genome similarities (to confirm or correct the taxonomy obtained with 16S rRNA sequencing). A pairwise ANIb calculation was carried out between all the possible combinations of the isolated strains and several strains of *B. subtilis*, *B. sonorensis* and *B. licheniformis* contained in the JSpeciesWS database [27]. The constructed dendrogram (Figure 1) clearly shows that the newly sequenced strains belong to the species *B. subtilis* (Z1), *B. licheniformis* (CG1) and *B. sonorensis* (AB6, AB7, X1 and X2).

Table 2 summarizes the predicted general features of each of the *Bacillus* genomes as well as the proteins that each of the strains might secrete according to RAST and SignalP-5.0 analyses.

### 3.3. Antimicrobial Assays

After the initial antimicrobial screening against *M. luteus* (see Figure 2), the six *Bacillus* isolates were also tested against other pathogenic bacteria, such as *Paenibacillus alvei* DSM29 (B), *Staphylococcus aureus* NCTC8325 (C) and *Mycobacterium smegmatis* MC2155. Antagonistic activity was only seen by *B. licheniformis* CG1 against *M. smegmatis* (Figure 2). Therefore, while *M. luteus* was susceptible to the six strains, *M. smegmatis* was only susceptible to the CG1 strain. As *Mycobacterium* species are quite resistant to antibiotic treatments, this result highlights the *B. licheniformis* CG1 isolate as an interesting antibiotic-producing strain.

### 3.4. Characterization of the Keratin-Degrading Activity of the Isolated Bacillus Strains

To study the keratin-degrading activity of the *Bacillus* isolates over keratin-rich wastes, the strains were assessed on their ability to grow on a medium with chicken feathers as the sole carbon and nitrogen source, as well as on their production of keratinase. The results showed in Figure 3 indicated that the strains with better degrading capabilities were *B. licheniformis* CG1 and *B. sonorensis* AB7. Both strains were able to grow and produce keratinases until the oxygen (O_2_) was depleted from the sealed bottes. Keratinase activity was more than 2.6 times higher in *B. licheniformis* CG1 in comparison to *B. sonorensis* AB7, and almost 16 times higher than the rest of isolates (see Table 3), which makes this strain a promising keratinase producer under minimal nutritional requirements.

### 3.5. Gene Cluster Associated with the Biosynthesis of Putative Antimicrobial Metabolites

Gene clusters associated with the synthesis of non-ribosomal peptide synthetases (NRPS), polyketide synthases (PKSs) and/or ribosomally synthesized and post-translationally modified peptides (RiPPs) were identified in most of the *Bacillus* isolates using both BAGEL3 and antiSMASH_v.6.0 software (see Appendix A). BAGEL3 was unable to identify NRPS, PKSs or RiPPs in AB7 and X2 strains. Nevertheless, antiSMASH_v.6.0 analysis predicted gene clusters for the production of lichenysin in all strains, with the exception of *B. subtilis* Z1. This metabolite has been shown to display antimicrobial activity against Gram-positive bacteria [33] and it could account for the inhibition of *M. luteus* growth by all *Bacillus* isolates tested. Surfactin [34], another predicted compound, would account for the antimicrobial activity displayed by *B. subtilis* Z1 (see Appendix A).

Interesting, *B. licheniformis* CG1, the only strain that inhibited the growth of the mycobacterial strain (see Figure 2), contained (in addition to the sequences with 100% similarity to lichenysin) another sequence with 100 % similarity to lichenicidin (see Figure 4). Lichenicidin is a lantibiotic compound with antimicrobial activity against a wide range of Gram-positive bacteria [35], which has been previously shown to be produced by another *B. licheniformis* strain, i.e., DSM13 [36]. This antibiotic might account for the antimicrobial activity against *M. smegmatis* observed by the CG1 strain; however, this hypothesis would require further testing.

### 3.6. B. licheniformis CG1 Keratinase Activity Detected across a Wide Range of pH

Factors such as microbial strain, medium composition, pH and temperature play major roles in the production of highly active keratinase enzyme. As we have shown, *B. licheniformis* CG1 displays the most interesting antimicrobial potential and the highest keratinase activity among the *Bacillus* isolates. Therefore, this strain would be a good candidate for biotechnological applications in feather valorization. For industrial approaches, it is important that the employed strain remains active across a range of pH, among other parameters. To study this, keratin degradation using smaller bottles and less volume of medium (20 mL instead of 200 mL) at pH values from 6 to 9 was performed. As is shown in Figure 5, the strain showed a similar keratinase activity with all pH values tested, reaching around 3000 units per mL in just 2–4 days, which might provide an advantage for further industrial applications.

## 4. Discussion

Famine and lack of resources are a problem in certain parts of the world, whereas waste management is a problem in developed countries, which results in environmental problems and diseases. In this sense, a more sustainable use of natural resources is of vital importance. This especially applies to those wastes that can serve as food, either for humans or for animals. A clear example is the feather residues generated by the poultry industry, which are usually discarded into the environment, generating ecological problems [37]. Biodegradation is a process carried out by organisms that transform waste (and other compounds) into novel products that can be of interest for the industry. Therefore, this natural process could play a key role in the creation of viable end-products using wastes of a different nature in a circular and environmentally friendly approach.

Feathers serve as an example of cheap waste with potential nutritional value [38]. Poultry farming produces thousands of tons of chicken feathers in every year [4]. This waste product cannot be digested by most animals; therefore, it has to be degraded in some form to be usable. Several microorganisms with keratinolytic activity have been reported in the last few decades [7]. In this study, we have isolated and characterized two *Bacillus* strains with significant keratinolytic activity. The highest keratinolytic activity was achieved by *B. licheniformis* CG1 (3800 U × mL^−1^). In previous work, *Bacillus altitudinis B12* displayed maximal keratinolytic activity of 1500 U × mL^−1^ [9]. In this study, *B. sonorensis* AB7 exhibited a comparatively similar magnitude of keratinolytic activity (1450 U × mL^−1^). This means that *B. licheniformis* CG1 shows a keratinolytic activity more than 2.6 times higher in comparison to the other two strains. Our results correlate well with previous studies. For example, in a study with *B. licheniformis* strain PWD-1, a similar magnitude of keratinolytic activity was reported (i.e., 3750 U × mL^−1^) [8]. Similarly, Hmidet et al. [39], after an in-depth optimization of the testing conditions, reported *B. licheniformis* strain NH1 with a maximal keratinase activity of 3960 U × mL^−1^. On the other hand, many studies have shown that the *Bacillus* strains analyzed (or other genera of bacteria) reached a maximal keratinolytic activity of 1000 U × mL^−1^ [40]. For example, *Bacillus pumilus* AR57 was found to be the most potent keratinase producer out of 39 isolates. However, the maximum keratinase activity reached by this strain was around 700 U × mL^−1^ [41], more than five times lower than *B. licheniformis* CG1, PWD-1 or NH1. In summary, our work corroborates previous studies that report that *B. licheniformis* species constitute one of the most important and effective keratin-degrading microorganisms discovered to date [8,39]. These observations highlight *B. licheniformis* species as promising candidates for feather degradation and keratinase production approaches. Keratinases are highly employed in different industrial applications, including animal nutrition, leather processing, detergent formulation and biofertilizer development [42]. In addition, as the bacterium grows on feathers as the sole carbon, nitrogen, sulfur and energy source, the utilization of feathers by these strains represents a method for replacing the commercial and more expensive substrates normally employed, therefore cheapening the cost of the culturing conditions. Thus, the utilization of feathers as substrate could result in a cost-effective process suitable for large-scale production of commercial enzymes produced by this species, such as those used in laundry detergent formulations [42]. The production of any other valuable metabolite that the bacterium could produce (either intrinsically or heterologously) would also be cheapened. Moreover, the absence of rapidly metabolizable carbohydrates in chicken feathers could also be beneficial for avoiding carbon catabolite repression [43], which is often observed in the production of extracellular proteases, including the well-known alkaline proteases produced by this bacterium [44].

This study has found that honeybee products, including honey and royal jelly, can contain *Bacillus* species. For centuries, it has been known that honey is an effective wound-healing agent and can even heal wounds that are unable to be healed by conventional treatments [45,46]. Studies into the antimicrobial activity of honey believed its effects were due to high osmolality of honey, and the presence of hydrogen peroxide [47]. However, more recent studies have shown that bacterial isolates from honey are able to exhibit antimicrobial activity against several foodborne pathogens, which suggests that the bacteria within honey contribute to its antimicrobial effects [48]. For example, it was shown that some *Bacillus* species, present in the gut of honeybees, were able to kill the pathogenic *Paenibacillus larvae*, which is the causative agent of foulbrood disease in bees, providing further indications that the bacteria within the honeybee microbiome may exhibit antimicrobial properties [49]. Actually, *Bacillus* species are an important source of antibiotics [50]. Some of these antimicrobial compounds are lipopeptides, which are produced by NRPS encoded in clusters on the bacterial genome [50]. Lipopeptides, that act through the disruption of membranes, are a particularly important source of antimicrobials, as it is less likely that resistance will develop towards them compared to other antibiotics. However, because of their toxicity, their use as therapeutics has been limited to mainly topical agents [51]. Well-known lipopeptides produced from *Bacillus* species include surfactin, fengysin and lichenysin, which have powerful antibacterial and antiviral properties [52]. Additionally, secondary metabolites of *Bacillus* can also be produced by PKS, also encoded within gene clusters on the bacterial genome. There are three categories of PKS: type I, type II and type III [53]. PKS are categorized in terms of structure; where the catalytic domains of type I PKS enzymes are within a single protein, the catalytic domains of type II and type III PKS enzymes are on separate proteins [54]. The possession of PKS genes is very common in *Bacillus* species, including in *B. subtilis* and *B. licheniformis*. In this study, and others, it has been shown that *B. licheniformis* is able to synthesize different antimicrobial compounds that can inhibit the growth of specific bacteria, which could be useful to tackle certain diseases associated with animals and plants in downstream applications with the strain [55]. The *B. licheniformis* CG1 isolate was predicted to have secondary metabolite clusters associated with the synthesis of lichenysin and lichenicidin, which is consistent with the literature [56,57]. Clusters for lichenysin were also present in the *B. sonorensis* isolates from this study. Of the six isolates found in the honeybee samples, four were identified as *B. sonorensis*. To our knowledge, this is the first report that describes the presence of *B. sonorensis* in honeybee products. Two *B. sonorensis* strains (X1 and X2) were isolated from royal jelly samples, which has never been reported before. The high nutritional value of royal jelly could make this substance as a good growth medium for these bacteria [58]. Alternatively, the presence of *Bacillus* strains in the royal jelly samples could be the result of a contamination, since bees mix royal jelly with pollen and nectar to produce worker jelly [59].

In summary, *B. licheniformis* has been defined, not only as one of the most important bacterial organisms employed in industry for enzyme production, but also as a promising probiotic organism for animal feed and as an efficient biofertilizer to prevent diseases and promote growth in plants [60]. Therefore, the *B. licheniformis* CG1 strain characterized in this study constitutes a promising candidate for use in different industrial applications based on keratin wastes utilization.

## Figures and Tables

**Figure 1 microorganisms-11-00456-f001:**
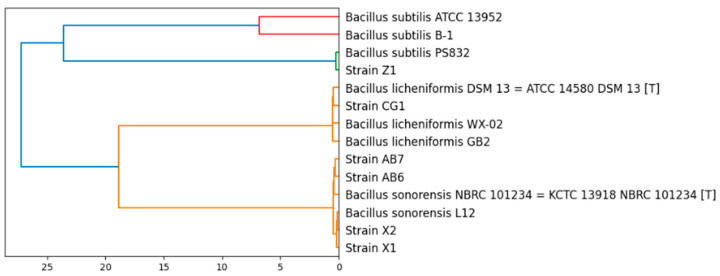
Phylogenetic relations of the isolated strains with strains of the species *B. subtilis*, *B. sonorensis* and *B. licheniformis*.

**Figure 2 microorganisms-11-00456-f002:**
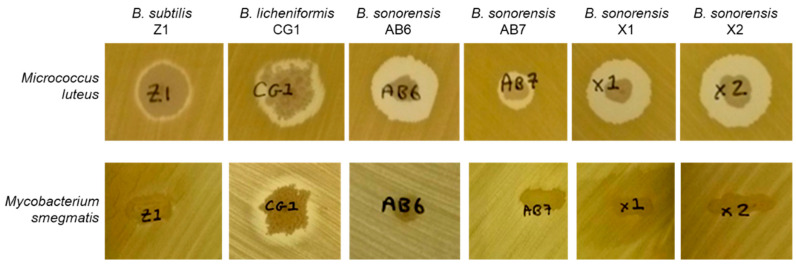
Antimicrobial assays of *Bacillus* isolates against *Micrococcus luteus* ATCC4698 and *Mycobacterium smegmatis* MC2155.

**Figure 3 microorganisms-11-00456-f003:**
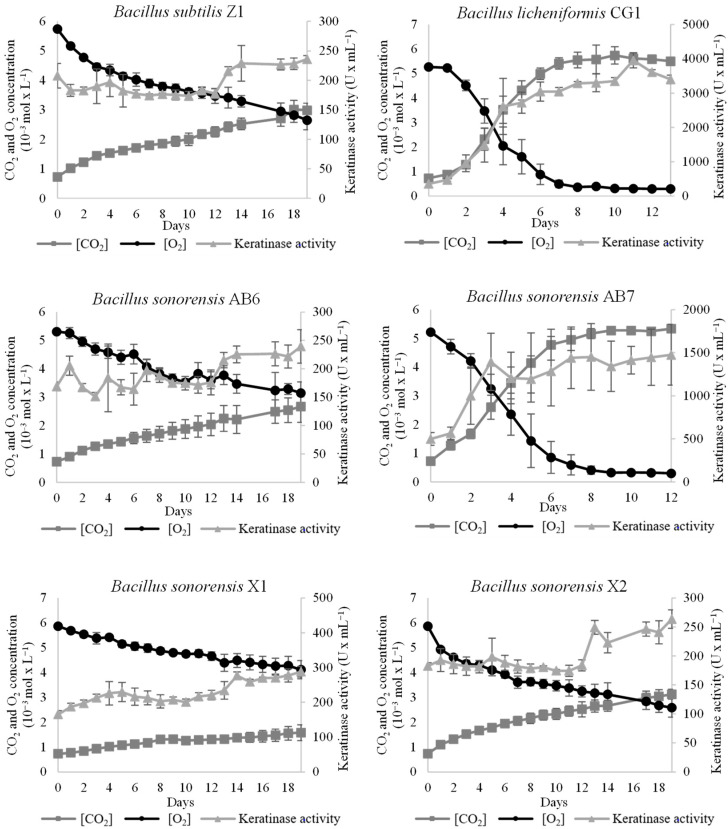
Growth and keratinase activity of the *Bacillus* isolates on chicken feathers as sole carbon and nitrogen source. Samples for growth (CO_2_ production and O_2_ consumption) and keratinase activity were collected every 24 h until O_2_ was depleted from the bottles or until when 20 days were reached. Vertical error bars correspond to the standard error of the mean of three replicated experiments.

**Figure 4 microorganisms-11-00456-f004:**
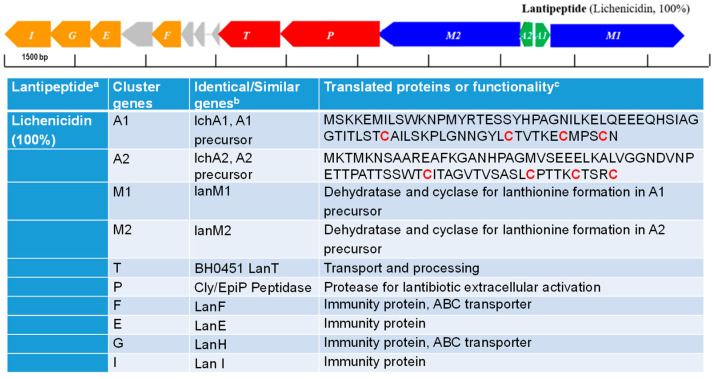
**Gene cluster associated with the synthesis of Lichenicidin identified in the genome of *B. licheniformis* CG1.** The color of the arrows indicate the nomenclature for genes encoding for: hypothetical precursor peptides (green arrows), posttranslational modification enzymes (blue arrows), transport/processing proteins (red arrows), immunity proteins (orange arrows) and hypothetical proteins (grey arrows). The hypothetical RiPPs are indicated next to each cluster, on the right-hand side, along with their corresponding identity alignment scores when compared to similar RiPPs (in brackets). In the Table: ^a^ percentage of similarity (indicated in brackets) with described lichenicidin; ^b^ Similarity with proteins reported previously; ^c^ cysteines involved in the lanthionine biosynthesis (indicated in red).

**Figure 5 microorganisms-11-00456-f005:**
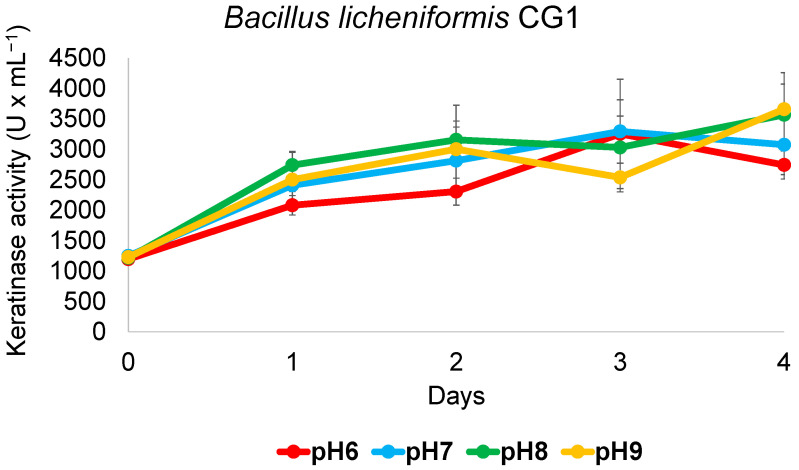
**Keratinase activity of *B. licheniformis* CG1 across a range of pH.** Cultures grown at 37 °C and 150 rpm for 4 days in 120 mL bottles hermetically closed showed similar keratinase activity in a medium adjusted to varying pH values and where chicken feathers were the sole carbon and nitrogen source. Vertical error bars correspond to the standard error of the mean of three replicated experiments.

**Table 1 microorganisms-11-00456-t001:** Strains isolated from honeybee samples in South England and characterized in this and previous studies.

Strain	Organism	Source	Location	BioSample	References
O29	*Lactobacillus kunkeei*	Bee pollen	Verney Junction, BUCKINGHAMSHIRE	SAMN11831833	[15]
E1	*Pediococcus acidilactici*	Raw honey	Saffron Walden, ESSEX	SAMN11831832	[15]
AD1	*Streptomyces drozdowiczii*	Raw honey	Woking, SURREY	SAMN20207146	[14]
AD2	*Streptomyces griseoaurantiacus*	Raw honey	Woking, SURREY	SAMN20207147	[14]
AN1	*Streptomyces* *albus*	Bee pollen	West Byfleet, SURREY	SAMN20207148	[14]
Z1	*Bacillus* *subtilis*	Raw honey	Woking, SURREY	SAMN31988539	This study
CG1	*Bacillus licheniformis*	Raw honey	Shere, SURREY	SAMN31988540	This study
AB6	*Bacillus sonorensis*	Raw honey	West Byfleet, SURREY	SAMN31988543	This study
AB7	*Bacillus sonorensis*	Raw honey	West Byfleet, SURREY	SAMN31988544	This study
X1	*Bacillus sonorensis*	Royal jelly	Woking, SURREY	SAMN31988541	This study
X2	*Bacillus sonorensis*	Royal jelly	Woking, SURREY	SAMN31988542	This study

**Table 2 microorganisms-11-00456-t002:** Data obtained from the genomes of the six *Bacillus* strains isolated in this study.

Strain	Genome Accession	Bases	Contigs	Encoded Proteins	Secreted Proteins
*Bacillus subtilis* Z1	JAPQWB000000000	4316935	305	4280	399
*Bacillus licheniformis* CG1	JAPQWC000000000	4228198	208	4252	407
*Bacillus sonorensis* AB6	JAPQWF000000000	4767461	595	4532	441
*Bacillus sonorensis* AB7	JAPQWG000000000	4580150	233	4440	437
*Bacillus sonorensis* X1	JAPQWD000000000	4903862	502	4770	447
*Bacillus sonorensis* X2	JAPQWE000000000	4809962	323	4807	445

**Table 3 microorganisms-11-00456-t003:** Summary of the antimicrobial and keratinolytic activities obtained for each of the strains.

Strain	Anti- *Micrococcus* Activity	Anti- *Mycobacterium* Activity	Max. Keratinase Activity (U/mL)
*Bacillus subtilis* Z1	Positive	Negative	240
*Bacillus licheniformis* CG1	Positive	Positive	3800
*Bacillus sonorensis* AB6	Positive	Negative	240
*Bacillus sonorensis* AB7	Positive	Negative	1450
*Bacillus sonorensis* X1	Positive	Negative	290
*Bacillus sonorensis* X2	Positive	Negative	260

## Data Availability

The sequenced genomes have been deposited at GenBank under the accession numbers: JAPQWB000000000 (Z1), JAPQWC000000000 (CG1), JAPQWD000000000 (X1), JAPQWE000000000 (X2), JAPQWF000000000 (AB6) and JAPQWG000000000 (AB7); BioProject: PRJNA907715.

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
