# Peer review of "Characterization of Bacillus Strains from Natural Honeybee Products with High Keratinolytic Activity and Antimicrobial Potential"

_microorganisms, 2023, doi:10.3390/microorganisms11020456_

Round 1

Reviewer 1 Report

The manuscript entitled “Isolation and characterization of Bacillus strains from natural honeybee products with high keratinolytic activity and antimicrobial potential” presented by Diego Martín-González and co-authors describes two Bacillus strains that are well characterized and of interest for biotechnological treatments of feathers. The uniqueness of these strains is combined keratinase and antimicrobial activity. The manuscript is well prepared and presented and I only have few minor questions/comments:

1) if I understood correctly, these strains were isolated previously (ref 20)? therefore, title needs a change

2) please all ‘minutes’ abbreviate to min

3) if possible, avoid handwriting in Fig 2

4) it would be great if quality of Figure 3 could be improved?

Author Response

The manuscript is well prepared and presented and I only have few minor questions/comments:

  • if I understood correctly, these strains were isolated previously (ref 20)? therefore, title needs a change

WE HAVE CHANGED THE TITLE

  • please all ‘minutes’ abbreviate to min

DONE

  • if possible, avoid handwriting in Fig 2

WE ARE SORRY BUT THOSE ARE THE ONLY PICTURES THAT WE HAVE FROM THE BIOASSAYS

  • it would be great if quality of Figure 3 could be improved?

PLEASE, NOTE THAT WE HAVE IMPROVED THE QUALITY OF THE FIGURE AND OTHER ASPECTS

Reviewer 2 Report

Why the material for bacterial isolation were honeybee products?

What were the indications that honeybee products are a good source of bacteria with keratinolytic properties. This requires some explanation.

The lack of numerical marking of the lines makes it difficult to prepare a review.

Identification of bacteria based on the V3-V4 region does not provide reliable results for bacterial species identification. The entire 16SrRNA gene should be sequenced.

2.5. It is not really a measurement of keratinase activity, but proteases that have the ability to hydrolyze caseine. This test can only be considered as a preliminary screening of bacteria with potential keratin degrading abilities.

It is not understood why antimicrobial activity is associated with keratinolytic activity. These are separate properties not related to each other in any way.

Fig.3. In the x-axis legend CO2 and O2 subscripts should be used, e.g.: CO2 and O2.

3.5. The results are only a hypothesis and have not been confirmed experimentally. The lack of methodology.

There are too many old references at work.

Author Response

  1. Why the material for bacterial isolation were honeybee products?

Honeybees collect nectar and pollen to manufacture their own food and, While producing these products, beehives accumulate a considerable number of microbes that might be beneficial for the insects. Very recent publications have reported that bacterial isolates from bees and honeybee products inhibit the growth of plant and bee pathogens. We have clarified (and expanded) these observations in both the introduction and discussion sections of the revised version of the manuscript.

  1. What were the indications that honeybee products are a good source of bacteria with keratinolytic properties. This requires some explanation.

There are no indicators that honeybee products are a good source of bacteria with keratinolytic properties. Actually, that was the aim of our screening. On the other hand, as we have mentioned in the previous question (and included in the revised version of the manuscript), honeybee products are a good source of bacteria with antimicrobial properties.

  1. The lack of numerical marking of the lines makes it difficult to prepare a review.

We are sorry for this. We used the template of the journal.

  1. Identification of bacteria based on the V3-V4 region does not provide reliable results for bacterial species identification. The entire 16SrRNA gene should be sequenced.

PLEASE, NOTE THAT WE HAVE NOT ONLY PERFORMED 16S ANALYSES BUT, To have a better insight about the taxonomy of the strain, we have ALSO performed genome sequencing analysis, WHICH allowed us to carry out a phylogenetic analysis using JSpeciesWS

  1. Section 2.5. It is not really a measurement of keratinase activity, but proteases that have the ability to hydrolyze caseine. This test can only be considered as a preliminary screening of bacteria with potential keratin degrading abilities.

it is clear the potential keratin degrading abilities of some of these strains since they can growth on feathers as sole carbon and nitrogen sources.

moreover, The fundament of the keratinase activity assay developped in this work is well described in the two papers included below:

  1. 10.1021/acsomega.0c05192
  2. 10.3791/899

The method employed in this paper is widely used in the literature for estimating the keratinase activity of many different bacterial strains. As an example, we include in this document 10 references in which author use this method for quantification of keratinase activity (see DOIs below):

  1. 10.4061/2010/132148
  2. 10.1371/journal.pone.0172712
  3. 10.1088/1755-1315/305/1/012084
  4. 10.1088/1755-1315/1059/1/012026
  5. 10.1007/s10562-021-03833-z
  6. 10.1007/s10295-010-0792-8
  7. 10.1007/s10529-013-1139-0
  8. 10.1007/BF02916414
  9. 10.1007/s00253-010-2534-2
  10. 10.1023/A:1023395409746
  11. It is not understood why antimicrobial activity is associated with keratinolytic activity. These are separate properties not related to each other in any way.

antimicrobial activity is not associated with keratinolytic activity. we have never stated this affirmation since both features are not related each other. Indeed, the aim of the work was to assess the ability of the strains for having both capabilities (since might be useful for downstream applications).

  1. 3. In the x-axis legend CO2 and O2 subscripts should be used, e.g.: CO2 and O2.

PLEASE, NOTE THAT WE CORRECTED THIS AND IMPROVED THE QUALITY OF THE FIGURE

  1. 5. The results are only a hypothesis and have not been confirmed experimentally. The lack of methodology.

The results are based on gene cluster predictions as it is clearly described in the manuscript. Actually, we use the conditional form in all the section and we clearly stated that the hypothesis would require further testing. In relation to the metodolgy, in the material and methods section, we clearly indicate the original references for both antiSMASH v.6.0 and BAGEL3 software.

  1. There are too many old references at work.

WE HAVE DONE OUR BEST FOR SELECTING THE REFERENCES IN THE STUDY TRYING TO INCLUDE ORIGINAL REPORTS. PLEASE, NOTE THAT 40 % OF THE REFERENCES THAT WE HAVE INCLUDED ARE FROM 2020 TO AFTERWARDS. IN ANY CASE, IN THE NEW TEXT THAT WE HAVE INCLUDED IN THE REVISED VERSION OF THE MANUSCRIPT SOME NEW REFERENCES OF RECENT WORK ABOUT THE HYPOTHESIS THAT BEEHIVES ARE A PROMISING SOURCE FOR THE IDENTIFICATION OF NOVEL BACTERIA WITH ANTIMICROBIAL POTENTIAL HAVE BEEN INCLUDED.

Round 2

Reviewer 2 Report

The Authors took into account all comments and made corrections according to the instructions. The work can be accepted in its present form.